# Physical activity and self-rated health during retirement transition: a multitrajectory analysis of concurrent changes among public sector employees

Roosa Lintuaho [1,2] Mikhail Saltychev [2] Jaana Pentti,[3,4] Jussi Vahtera [3,5] Sari Stenholm [3,5]

¹Department of Emergency Medicine, TYKS Turku University Hospital, Turku, Finland
²Department of Physical and Rehabilitation Medicine, TYKS Turku University Hospital, Turku, Finland
³Department of Public Health, University of Turku, Turku, Finland
⁴Clinicum, Faculty of Medicine, University of Helsinki, Helsinki, Finland
⁵Centre for Population Health Research, University of Turku, Turku, Finland

**Correspondence to**
Dr Roosa Lintuaho;
roemti@utu.fi

## ABSTRACT

**Objectives** The aim of the study was to evaluate concurrent changes in physical activity and self-rated health during retirement transition over 4 years by multivariate trajectory analysis and to examine whether sociodemographic and lifestyle factors predict the probability of being classified to a certain subgroup of observed changes.

**Design** Prospective cohort study.

**Setting** Public sector employees.

**Participants** 3550 participants of the Finnish Retirement and Aging study.

**Primary and secondary outcome measures** Participants estimated on a yearly questionnaire their weekly hours of different types of activities converted to metabolic equivalent of task-hour/week. Self-rated health was assessed on a 5-point Likert-like scale from poor to excellent and dichotomised as suboptimal and optimal. Multivariate trajectory analysis was used to distinguish different subgroups of trajectories. Multinomial regression analysis was used to describe the associations between covariates and the probability of being classified to a certain trajectory group.

**Results** Three trajectory groups were identified, all displaying increasing activity during retirement with a simultaneous decrease in perceived suboptimal health. Physical activity peaked at 18 months after retirement and then slightly decreased, except for initially highly physically active participants (9%) with good self-rated health, who demonstrated a constant high level of physical activity. Male gender, professional occupation, being married or cohabiting, body mass index <30 kg/m², not smoking and using alcohol below risk levels were associated with higher physical activity and better self-rated health.

**Conclusion** Changes in physical activity and perceived health during retirement transition were interconnected. Both were improved during retirement transition, but the change was temporary. Longer follow-up studies are required to assess the changes over a longer period after retirement.

## INTRODUCTION

Retirement may change many daily routines, for example, diet, sleeping rhythm, psychological well-being, as well as the types

## STRENGTHS AND LIMITATIONS OF THIS STUDY

⇒ A data-driven approach to finding subgroups with different trajectory paths.
⇒ Large sample size, representative of the public sector workforce.
⇒ Repeated measurements of self-rated health and physical activity.
⇒ The self-reported nature of the variables might lead to information bias.
⇒ Participants were mostly retiring due to old age, leaving out those retiring due to health reasons, who might get different results.

and intensity of leisure-time activities or hobbies.[1–4] Such a comprehensive change may also affect health perception.

Several studies have reported increasing physical activity during the retirement transition.[1 5–12] According to multiple follow-up studies, this increase might be temporary.[6 10 13] The type of retirement affects the change in physical activity—retiring due to health reasons has been associated with lesser increase or even decreasing physical activity after retirement.[7 9] Gender, work characteristics and socioeconomic status may also moderate these changes. Studies by Touvier *et al*[14] and Lahti *et al*[7] have reported greater increase in physical activity during retirement transition among men, whereas study by Sjösten *et al*[5] has shown greater increase in physical activity among retiring women. Retiring from non-manual work seems to be beneficial regarding physical activity comparing with retiring from manual jobs.[6 15 16] Systematic reviews have also reported the effect of socioeconomic status on the intensity of physical activity after retirement—higher status has been associated with improved physical activity.[1 17] Some sex-related differences have been reported concerning the effect of occupational status—women retiring from manual

work have showed decreased accelerometer-measured physical activity, while physical activity has not changed among women retiring from non-manual work.[16] Respectively, men retiring from non-manual work had increased physical activity, while those retiring from manual work have not demonstrated any change.[16]

Previous knowledge on the change in self-rated health during retirement transition has been inconsistent. In a French study among employees, who retired at relatively young age (54 years on average), perceived health has improved after retirement, especially for those with poorer working conditions and health problems prior to retirement.[18] In a systematic review, mental health has often substantially improved after retirement. Instead, the change in self-rated health has been less clear as different studies have reported improvement, deterioration or no change in self-rated health.[19] A meta-analysis by van der Heide *et al* has reported either deterioration or improvement in self-rated health depending on which set of original studies has been pooled.[19] It seems that retirement due to health reasons is associated with greater positive changes in perceived health compared with those who retired due to old age.[19] Worsening physical health has been associated with lower occupational status, manual work and higher prevalence of chronic conditions.[20] In another study, job strain and physically demanding work have been associated with suboptimal health prior to retirement. In turn, sustained good health has been associated with normal weight, non-smoking and higher physical activity.[21]

Previous research has found associations between physical activity and self-rated health.[21 22] Thus, it can be assumed that changes in physical activity and self-rated health may follow the same patterns around the retirement transition. Simultaneous changes on the subject have not been studied before.

The aim of this study was to evaluate concurrent changes in physical activity and self-rated health during retirement transition among Finnish public sector employees by using a multitrajectory analysis to identify subgroups with different trajectory paths during this transition.

## METHODS
### Study population
The Finnish Retirement and Aging Study (FIREA) is an ongoing longitudinal cohort study, which follows ageing workers from final years in working life to full-time retirement and until old age.[23] The eligible population for the FIREA included all public sector employees whose individual retirement date was between 2014 and 2019 and who were working in the year 2012 in one of 27 municipalities, 9 cities or 5 hospital districts included in the study. Municipalities and cities, involved in the study, are responsible for school education, daycare centres, social services, libraries, maintaining roads and primary healthcare services, among others. Respectively, hospital districts, involved in the study, are responsible

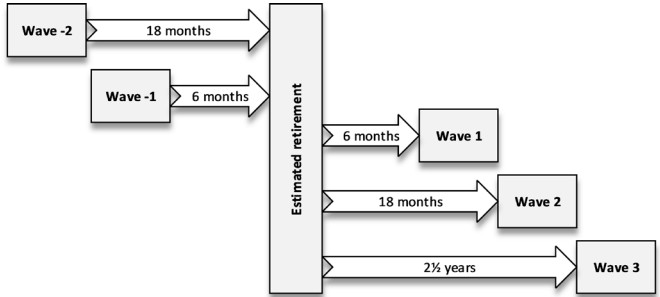

**Figure 1** Study waves and time in relation to retirement.

for providing secondary healthcare services. Thus, the eligible population covered a wide variety of occupations.

The participants were first contacted 18 months prior to their estimated retirement date, which was obtained from the register kept by the pension provider for public sector employees (Keva). The actual retirement date was self-reported. The participants received a questionnaire, which has been thereafter sent annually, at least four times, expanding the follow-up 2.5 years after retirement. Participants who had provided information regarding physical activity and self-rated health at least two consecutive times, one right before retirement and one after, were included in the current study (n=3550). There were two possible survey waves before the retirement, waves −2 and −1, and three possible waves after the retirement, waves 1, 2 and 3 (figure 1).

### Measurement of physical activity and self-rated health
Physical activity was self-reported at each study wave. The participants were asked to estimate their average weekly hours of leisure-time physical activity (including active travel to work) in activities comparable to walking, brisk walking, jogging or running.[24] The level of physical activity was then converted into metabolic equivalent of task (MET), which describes the amount of consumed energy comparing to resting. One MET unit of 3.5 mL/kg/min corresponds to oxygen consumption while sitting at rest. Weekly physical activity was expressed as MET-hour/week. To help the interpretation of the results, following categorisation was used: low (<14 MET-hour/week), moderate (14 to <30 MET-hour/week) or high (≥30 MET-hour/week) physical activity.[24 25] This categorisation was chosen since physical activity lower than 14 MET-hour/week (approximately the equivalent of 140 min of brisk walking) has been reported to be associated with higher risk of cardiovascular disease[26] and the activity level of 30 MET-hour/week (the equivalent of 300 min of brisk walking) has been shown to be needed for weight management.[27]

Self-reported health was inquired at each study wave. Participants were asked to rate their health on a five-point Likert-like scale:1-good, 2-rather good, 3-average, 4-rather poor or 5-poor. For the purpose of analysis, the result was dichotomised as 3–5 as 'suboptimal health' vs 1–2 as 'optimal health'.

## Independent variables

Age was defined in years. Marital status was dichotomised as 'married or cohabiting' versus 'single'. Occupational titles were obtained from the register of pension provider (Keva), and they were coded according to the International Standard Classification of Occupations (ISCO) and dichotomised as 'professionals' (ISCO major groups 1–4) versus 'service and manual workers' (ISCO major groups 5–9). Body mass index (BMI) was calculated as weight/height$^2$ according to self-reported body height and weight. BMI was dichotomised as 'normal and overweight' (BMI<30 kg/m$^2$) vs 'obesity' (≥30 kg/m$^2$). Smoking was dichotomised as 'current smokers' versus 'never-smokers and former smokers'. Alcohol consumption was obtained from the survey as units consumed weekly and converted into grams of pure alcohol per week (g/week). Amount of >288 g/week for men and >192 g/week for women were considered a cut-off for excess alcohol consumption and dichotomised as 'risk users' versus 'non-risk users'. Information on covariates was obtained at the last available wave before retirement, wave −1.

## Statistical analysis

The covariates were reported as means and SD or as absolute numbers and percentage, when appropriate.

Group-based multitrajectory analysis (GBTA) was used to investigate the developmental trajectories (a course of outcome over time) of physical activity and self-reported health. The GBTA is a form of finite mixture modelling for analysing longitudinal repeated measures data.[28] It is a data-driven form of analysis which is able to distinguish and describe subpopulations (clusters) existing within a studied population, while conventional statistics only show the average outcome over time. The trajectories of such subpopulations may differ substantially from each other and from the average trajectory observed in the entire population. The censored (known also as 'regular') normal model of GBTA was used. The goodness-of-model fit was judged by running the procedure several times with a number of clusters starting from one up to four. The Bayesian information criterion (BIC), Akaike information criterion (AIC) and average posterior probability (APP) were used as criteria to confirm the goodness of fit. Linear, quadratic and cubic regression models were tested and cubic model was retained for using in the analysis. The cut-off for the smallest group was set at >5% of the entire cohort. Physical activity was treated as a continuous variable. Self-rated health was dichotomised in the analysis as suboptimal (yes/no). For sensitivity analysis, self-rated health was also used as a continuous variable.

Multinomial regression analysis was used to describe the associations of demographic and lifestyle factors and probability of being classified into a particular cluster. The results were presented as risk ratios (RRs) and their 95% CIs. The RRs were adjusted for age and gender, except for the RR for gender which was only adjusted for age.

The analyses were performed using Stata/IC Statistical Software: Release V.17. (StataCorp). The additional freely available Stata module 'traj' was required to conduct group-based trajectory analysis.[29]

## Patient and public involvement

None.

## RESULTS

The characteristics of the study sample prior to retirement (study wave −1) are presented in table 1. Of the respondents, 83% were women, 31% were single and 65% were professional workers. Mean age before the retirement was 63.4 (SD 1.4) years. Only 9% were currently smoking, 2% consumed alcohol over risk limits and 28% had a BMI≥30 kg/m$^2$. Of the respondents, 24% rated their health as suboptimal and 22% reported low, 18% moderate and 60% high physical activity.

**Table 1** Preretirement characteristics of the study population (wave −1)

| Variable | n | % |
|---|---|---|
| Gender | | |
| Women | 2949 | 83 |
| Men | 601 | 17 |
| Marital status | | |
| Single | 1088 | 31 |
| Cohabiting | 2462 | 69 |
| ISCO class | | |
| Professional | 2283 | 65 |
| Manual or service worker | 1239 | 35 |
| BMI | | |
| BMI<30 kg/m$^2$ | 2569 | 72 |
| BMI≥30 kg/m$^2$ | 981 | 28 |
| Smoking | | |
| Not currently | 3192 | 91 |
| Currently smoking | 305 | 8.72 |
| Alcohol consumption | | |
| No risk use | 3457 | 98 |
| Risk use | 87 | 2 |
| Suboptimal self-rated health | | |
| No | 2690 | 76 |
| Yes | 853 | 24 |
| Physical activity | | |
| Low (<14 MET-hour/week) | 770 | 22 |
| Moderate (14 to <30 MET-hour/week) | 638 | 18 |
| High (≥30 MET-hour/week) | 2142 | 60 |
| | Mean | SD |
| Age | 63.4 | 1.4 |

BMI, body mass index; ISCO, International Standard Classification of Occupations; MET, metabolic equivalent of task.

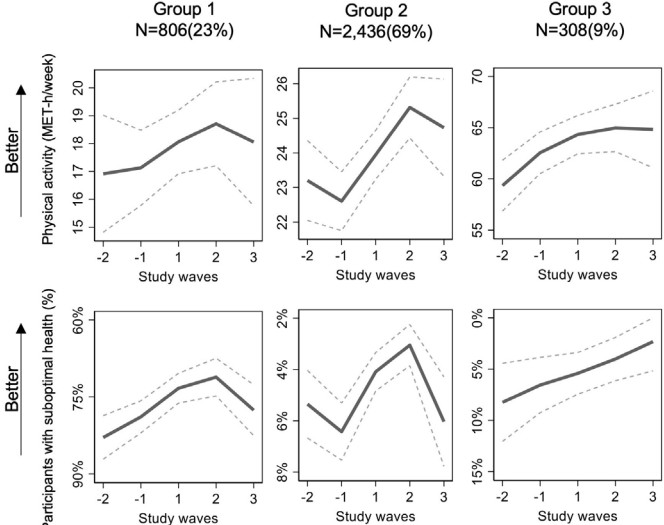

**Figure 2** Trajectories of physical activity and self-rated health during retirement transition. The scale is different in each group. MET, metabolic equivalent of task.

## Trajectory groups

Three trajectories of concurrent changes in physical activity and self-rated health were identified (figure 2). The three-cluster model was chosen, since the smallest group in the next, four-cluster model, fell below the predefined cut-off of 5% (table 2). The smallest APP was sufficient 0.93 for the three-cluster model. Compared with the two-cluster model, a three-cluster model demonstrated BIC and AIC values closer to 0. The results are also presented using the same scale for a y-axis in online supplemental file 1. The estimates of physical activity and self-rated health by trajectory group at each wave are presented in online supplemental file 2.

### Group #1: moderate physical activity and suboptimal self-rated health (23%)

Physical activity was at a moderate level of 17.1 MET-hour/week 6 months before retirement. First, it slightly increased from final working years until 18 months after retirement up to 18.1 MET-hour/week and thereafter slightly decreased. The trajectory of self-rated health was mirroring the changes in physical activity: 79% reported suboptimal health 6 months prior to retirement

improving down to 71% 18 months after the retirement and then slightly worsening again.

### Group #2: moderate physical activity and good self-rated health (69%)

Physical activity was at a moderate level of 22.6 MET-hour/week 6 months before retirement and increased during retirement transition all the way to 18 months after the retirement to 24.4. MET-hour/week. After that, physical activity showed a mild decrease staying, however, higher than the initial level. In this group, only 6% of the respondent reported suboptimal health 6 months before the retirement and decreasing down to 3% on the way to 18 months after the retirement. After that time point, the percentage of people with suboptimal health grew again up to 6%.

### Group #3: high physical activity and good self-rated health (9%)

This smallest group demonstrated a high level of physical activity and good self-rated health during the entire follow-up. The intensity of physical activity increased from 62.6 MET-hour/week 6 months before retirement up to 65 MET-hour/week 18 months after retirement. There was no decline in physical activity, unlike in the other two groups. Respectively, the portion of respondents with suboptimal health decreased from 7% to 4%.

### Sensitivity analysis

The trajectory solution was similar when the analysis was rerun treating self-rated health as a continuous variable instead of binary variable (online supplemental file 3). The respective groups were:

Group #1: moderate physical activity and suboptimal self-rated health (31%).

Group #2: moderate physical activity and good self-rated health (61%).

Group #3: increasing high physical activity and good self-rated health (8%).

### Associations between sociodemographic and health behavioural factors and trajectory groups

Compared with group #1 (moderate physical activity and suboptimal self-rated health), in groups #2(moderate physical activity and good self-rated health) and #3

**Table 2** Goodness of fit of group-based trajectory analysis models

| No of clusters | Shape of trajectory | Smallest group | | BIC | AIC | Smallest APP |
| --- | --- | --- | --- | --- | --- | --- |
| | | N | % | | | |
| 1 | Cubic | 3550 | 100 | 66 605 | 66 568 | 1.0 |
| 2 | Cubic | 861 | 24 | 64 626 | 64 552 | 0.94 |
| **3** | **Cubic** | **308** | **9** | **62 956** | **62 845** | **0.93** |
| 4 | Cubic | 64 | 2 | 62 434 | 62 286 | 0.89 |

The chosen model is shown in bold. Self-rated health treated as a binary variable.
AIC, Akaike information criterion; APP, smallest average posterior probability; BIC, Bayesian information criterion.

(increasing physical activity and improving self-rated health), there were more married or cohabiting persons (RR 1.23, 95% CI 1.04 to 1.47/RR 1.48, 95% CI 1.10 to 1.99), fewer manual workers (RR 0.77, 95% CI 0.65 to 0.91/RR 0.66, 95% CI 0.50 to 0.88), fewer overweight or obese participants (RR 0.35, 95% CI 0.30 to 042/RR 0.13, 95% CI 0.09 to 0.19), fewer smokers (RR 0.68, 95% CI 0.52 to 0.88/RR 0.31, 95% CI 0.17 to 0.58) and fewer risk users of alcohol (RR 0.60, 95% CI 0.38 to 0.96/RR 0.23, 95% CI 0.07 to 0.77) (table 3).

When comparing the most active and healthy groups #2 and #3, there were fewer women (RR 0.63, 95% CI 0.47 to 0.84), fewer persons with BMI>30 kg/m$^2$ (RR 0.36, 95% CI 0.25 to 0.53) and smokers (RR 0.46, 95% CI 0.25 to 0.83) in group #3 than in group #2 (table 3).

## DISCUSSION

In this prospective cohort study among 3550 public sector employees, three trajectory groups of concurrent changes in physical activity and self-rated health were identified. Overall, in all three groups, physical activity and self-rated health were clearly interconnected; trajectories for both parameters were mirroring each other—increase in physical activity was accompanied by improving health perception and vice versa. Higher physical activity was clearly associated with better self-rated health. These findings mirror previous reports on association between higher physical activity and better self-rated health.[22 30 31] However, concurrent trajectories of changes in these two parameters have not been studied before.

Overall, physical activity and self-rated health increased simultaneously during retirement transition. For 91% of the participants, this increase was only temporary. In regard of physical activity, this has been seen in previous research as well.[6 10 13] Only in a small proportion of participants, who were initially highly active and in good health, the improvement was maintained after the retirement transition throughout the entire follow-up. In this group, there were more men, professional workers and participants with BMI<30 kg/m$^2$ than in other groups, which is in line with previous findings; higher physical activity after retirement has been associated with higher socioeconomical status,[1 12 17] normal weight[12] and in some studies, male sex.[7 14]

Some previous studies have observed different paths of self-rated health. A study from New Zealand, identified three trajectories of self-rated health: declining, recovering and one with rather stable self-rated health that has shown only a minor decrease after retirement.[20] In a Finnish study, four trajectories around retirement transition were observed: two with sustained level of health, either good or suboptimal, one with improving self-rated health and one with declining health.[21] The difference in these results compared with current could be due to the difference in study samples, but also due to the multitrajectory setting of the current study, in which changes in

physical activity are just as important when defining the trajectories.

Compared with the group with moderate activity and poor self-rated health, there were more professional workers, persons with normal weight, non-smokers, cohabiting participants and participants who consume alcohol within moderation in the other two groups with higher activity and better self-rated health. These findings are in line with previous work; higher physical activity after retirement has been associated with retiring from professional work and higher socioeconomical status[1 6 17] and lower BMI.[12] Poorer self-rated health has also been found to be associated with lower socioeconomic status, overweight and lower physical activity,[21 32] and good self-rated health has been associated with normal weight, non-smoking and normal level of physical activity.[21]

Although there was improvement in all groups regarding physical activity and self-rated health, the improvement in self-rated health was most prominent in the group with lowest activity and poorer self-rated health; the proportion of participants with suboptimal health decreased by 12%, whereas in other groups the change was 3%–6% during follow-up (although due to small percentage of individuals experiencing suboptimal health in these groups, such an improvement was not possible). In this group, there were more single people, manual workers, participants with obesity, current smokers and participants with risk use of alcohol. This has been seen in other studies as well; in the New Zealand study, retirement has been most beneficial in regards of self-rated health to those with manual occupations, low to medium socioeconomic statuses and multiple chronic conditions.[20] On the other hand, some studies have found retirement to be associated with declining self-rated health for people with higher socioeconomic status, which was not the case in the current study.[19 20]

### Strengths and limitations

This study was conducted on a large cohort, representative of the public sector workforce. Strengths of this study also include a data-driven approach to defining clusters and repeated measurements of physical activity and self-rated health.

There are some limitations as well, which should be considered. The participants were, due to the inclusion criteria, mostly retiring due to old age. Since most of the participants could keep on working until their retirement age, they were probably healthier and in better shape than those who retired earlier due to health reasons. Although the sample is representative of public sector employees in Finland, the sample was predominated by women (83%) and professional workers (65%), which affects the generalisability of these findings into the general population. This study only assessed leisure-time and active travel to work, leaving out work-related physical activity. The self-reported nature of the estimates could lead to information bias. Self-rated health is, although, a widely used measurement perceived as a valid indicator of sustaining

**Table 3** Risk ratios (RR) and their 95% CIs of being classified to a certain trajectory group

| Variable/group | Group #1 RR | 95% CI | Group #2 RR | 95% CI | Group #3 RR | 95% CI |
|---|---|---|---|---|---|---|
| Women vs men | | | | | | |
| Group #1: moderate physical activity and suboptimal self-rated health | 1 | | 0.82 | 0.66 to 1.01 | 1.29 | 0.94 to 1.78 |
| Group #2: moderate physical activity and good self-rated health | 1.23 | 0.99 to 1.51 | 1 | | **1.59** | **1.19 to 2.11** |
| Group #3: increasing physical activity and improving self-rated health | 0.77 | 0.56 to 1.07 | **0.63** | **0.47 to 0.84** | 1 | |
| Co-habiting vs single | | | | | | |
| Group #1: moderate physical activity and suboptimal self-rated health | 1 | | **0.81** | **0.68 to 0.96** | **0.68** | **0.50 to 0.91** |
| Group #2: moderate physical activity and good self-rated health | **1.23** | **1.04 to 1.47** | 1 | | 0.84 | 0.64 to 1.10 |
| Group #3: increasing physical activity and improving self-rated health | **1.48** | **1.10 to 1.99** | 1.20 | 0.91 to 1.57 | 1 | |
| Manual workers vs professional workers | | | | | | |
| Group #1: moderate physical activity and suboptimal self-rated health | 1 | | **1.30** | **1.10 to 1.53** | **1.52** | **1.14 to 2.02** |
| Group #2: moderate physical activity and good self-rated health | **0.77** | **0.65 to 0.91** | 1 | | 1.17 | 0.90 to 1.52 |
| Group #3: increasing physical activity and improving self-rated health | **0.66** | **0.50 to 0.88** | 0.86 | 0.66 to 1.11 | 1 | |
| BMI≥30 vs BMI<30 | | | | | | |
| Group #1: moderate physical activity and suboptimal self-rated health | 1 | | 2.82 | 2.38 to 3.33 | 4.05 | 5.23 to 11.58 |
| Group #2: moderate physical activity and good self-rated health | **0.35** | **0.30 to 0.42** | 1 | | **2.76** | **1.88 to 4.05** |
| Group #3: increasing physical activity and improving self-rated health | **0.13** | **0.09 to 0.19** | **0.36** | **0.25 to 0.53** | 1 | |
| Current smokers vs former/never smokers | | | | | | |
| Group #1: moderate physical activity and suboptimal self-rated health | 1 | | **1.47** | **1.13 to 1.90** | **3.19** | **1.72 to 5.93** |
| Group #2: moderate physical activity and good self-rated health | **0.68** | **0.52 to 0.88** | 1 | | **2.18** | **1.20 to 3.95** |
| Group #3: increasing physical activity and improving self-rated health | **0.31** | **0.17 to 0.58** | **0.46** | **0.25 to 0.83** | 1 | |
| Risk users of alcohol vs non-risk users | | | | | | |
| Group #1: moderate physical activity and suboptimal self-rated health | 1 | | **1.66** | **1.05 to 2.62** | **4.33** | **1.31 to 14.39** |
| Group #2: moderate physical activity and good self-rated health | **0.60** | **0.38 to 0.96** | 1 | | 2.62 | 0.81 to 8.46 |
| Group #3: increasing physical activity and improving self-rated health | **0.23** | **0.07 to 0.77** | 0.38 | 0.12 to 1.24 | 1 | |

RRs are adjusted for age and gender (only for age for gender RR). Statistically significant findings are shown in bold.

well-being. It is an independent predictor of mortality,[33] as well as increased use of prescribed medications and healthcare and social services.[34] To minimise the risk of bias due to the measurement used, the analyses were conducted separately with self-rated health treated as a dichotomised variable and as a continuous variable.

Further research is needed on interventions on how to maintain the positive trend in self-rated health and physical activity in the years following retirement.

## CONCLUSIONS

Physical activity was found to increase, and self-rated health improved during retirement transition, although this change was temporary for most participants. Changes in physical activity and self-rated health seemed to be interconnected. Lower physical activity, being single, manual work, obesity, current smoking and risk use of alcohol were associated with poorer self-rated health and lower physical activity.

**Contributors** All the authors (RL, MS, JP, JV and SS) substantially contributed to the conception and design of the work, the interpretation of the results and revising it critically for important intellectual content. SS and JV were responsible for the acquisition of data for the work. MS and JP were responsible for the statistical analysis. RL was responsible for drafting the work. SS was the guarantor. All the authors (RL, MS, JP, JV, SS) have finally approved the version to be published and they are agreed to be accountable for all aspects of the work in ensuring that questions related to the accuracy or integrity of any part of the work are appropriately investigated and resolved.

**Funding** This study was supported by funding granted by the Academy of Finland (321409 and 329240 to JV, 286294, 319246, 294154, 332030 to SS), Finnish Ministry of Education and Culture (to SS); Juho Vainio Foundation (to SS) and Competitive State Research Financing of the Expert Responsibility Area of the Turku University Hospital (to RL and SS).

**Competing interests** None declared.

**Patient and public involvement** Patients and/or the public were not involved in the design, or conduct, or reporting, or dissemination plans of this research.

**Patient consent for publication** Not applicable.

**Ethics approval** This study involves human participants and the FIREA has been approved by the Ethics Committee of Hospital District of Southwest Finland. Participants gave informed consent to participate in the study before taking part.

**Provenance and peer review** Not commissioned; externally peer reviewed.

**Data availability statement** Data are available on reasonable request. Anonymised partial datasets of the Finnish Retirement and Aging Study are available by application with bona fide researchers with an established scientific record and bona fide organisations. For more information, please contact, SS ( firea@utu.fi).

**ORCID iDs**
Roosa Lintuaho http://orcid.org/0000-0002-7781-8922
Mikhail Saltychev http://orcid.org/0000-0003-1269-4743
Jussi Vahtera http://orcid.org/0000-0002-6036-061X
Sari Stenholm http://orcid.org/0000-0001-7560-0930

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
