## [Reviewer comments · BMJ Open]

ARTICLE DETAILS

TITLE (PROVISIONAL)	Physical activity and self-rated health during retirement transition – a multi-trajectory analysis of concurrent changes among public sector employees
AUTHORS	Lintuaho, Roosa; Saltychev, Mikhail; Pentti, Jaana; Vahtera, Jussi; Stenholm, Sari

VERSION 1 – REVIEW

REVIEWER	Hess , Moritz Hochschule Niederrhein University of Applied Sciences
REVIEW RETURNED	21-Apr-2023

GENERAL COMMENTS	I had the honor to review the paper Self-rated health and physical activity during retirement transition – a multi-trajectory analysis of concurrent changes. In the paper physical activity and self-rated health during retirement transition were explored. Overall I did like the paper. The idea was clear and the paper well written. The methods were innovative and provided interesting results. I have only one point of concern. As mentioned in the limitations section the group investigated – mainly women in the Finish public sector – is a highly selective group and the results can not be generalized to the other groups. This should be made clear not only in the discussion but in the whole manuscript. Starting with the title: this could be changed to for example “Self-rated health and physical activity during retirement transition – a multi-trajectory analysis of concurrent changes in the Finish public sector”. The limitation of a selective sample could also be made prominent in the discussion and mentioned on the introduction. Apart from that I like the paper very much.
---

REVIEWER	Brainard, Julii University of East Anglia, Norwich Medical School
REVIEW RETURNED	26-Apr-2023

GENERAL COMMENTS	Thanks for a well written article. My comments amount to only minor suggestions. I'm not familiar with GBTA and appreciate the clear explanations. Generally the writing on this was very clear which is much appreciated. MOST IMPORTANT COMMENTS: Please in abstract state the follow up period (generally 2-3 years, I think). Although PA does seem to decline for > 90% of participants, the actual change from peak (in absolute terms) is small. Is it possible to change the message to reflect that the decline is small in the monitored period. Also could abstract state that observed peak
---

seemed to be about 18m post-retirement for most. Without these revisions, the word “temporary” is vague, not defined in the abstract & elsewhere.

Supplemental file 3 has variable vertical scales on charts ; trying to same vertical scales on all charts in Supplemental 2 figures makes them all look like flat lines. I would personally prefer either 1) limited scales showing just the range needed or 2) change from baseline which could then be all same scale on Suppl Fig 2 charts. Flat lines are really not informative, though, and don't help explain the cubic model forms. It could be argued that changes from 3% to 6% developing suboptimal health is really quite a small change in absolute terms, even if it was double.

Supplemental files 2/3 : are those dashed lines 90% ranges or IQRs or standard deviations ? Please label dashed lines on figures. Definitely did want variance info on the group MET-h.... also , in RESULTS, I think you are talking about “means” not medians as central estimates, with statements like “moderate level of 22.6” : but not sure. Is that median or mean?

Although there is a decline in PA (and it's good you linked it to self-described health status) for 91%, the decline is actually very small. Surely for public health messaging and ideally supporting people to keep up PA, the rate of decline is important. Calling the increase in PA “temporary” is a bit simplistic because it sounds like a greater decline in PA than was actually observed.

METHODS

Please can text reiterate that there were 2.5 years of post-retirement activity data for all participants. This helps to define “temporary”.

Public sector can cover a wide range of occupational groups. In this study, does public sector include health care workers, teachers, police officers?

The explanation of MET-h/week is very clear, thank you.

The abstract suggests that being normal/overweight is associated with higher PA but the detail is that the relevant category is really “not obese” : unless you are saying not a single participant was underweight?

Sentence that would be more grammatically correct written as “Amounts of > 288 g/week for men and > 192 g/week for women were considered as cut-offs for excess alcohol...”

Penultimate paragraph in methods ends in a sentence (or sentence fragment?) that I didn't understand, twice saying RRs were adjusted for age and gender; is this just unnecessarily repetitive?

I wasn't sure why mention the module being available for other stats software: It seemed like extraneous detail wrt to this study.

RESULTS

Was the achieved sample representative of the eligible respondents? About as female, distributed in about same types of occupations?

	Group#3 results: text should make clear that there was not a decline from peak PA time after 18m point, unlike declines after 18m point for Group#1 & Group#2. OTHER SUGGESTED CHANGES: Quite often the text uses perfect past tense when simple past tense would be plainer English. For instance, “has reported” could be “reported”, “set of original studies was pooled”, “participants received a questionnaire which was thereafter sent annually” etc. “compared with those retired due to” would be better “compared with those who retired due to” .. (double full stop) just before reference 24 Physical activity while commuting is called active travel, conventionally; I wondered if you should use that phrase; leisure PA is a different type of PA Paragraph about Associations with sociodemographic & Health behavioural factors.... Throughout this paragraph need to say “fewer” not “less”; fewer for countable items. https://style.mla.org/fewer-versus-less/ I don’t think New Zealandean is a word. At least, I think simply “New Zealand” would be better in the adjective form. Recommend insert “physical” between work-related & activity, penultimate paragr in discussion
--	---

VERSION 1 – AUTHOR RESPONSE

Responses to comments made by reviewer 1:

Comment 1

As mentioned in the limitations section the group investigated – mainly women in the Finish public sector – is a highly selective group and the results can not be generalized to the other groups. This should be made clear not only in the discussion but in the whole manuscript. Starting with the title: this could be changed to for example “Self-rated health and physical activity during retirement transition – a multi-trajectory analysis of concurrent changes in the Finish public sector”. The limitation of a selective sample could also be made prominent in the discussion and mentioned on the introduction.

Response 1

We have rephrased the title as follows:

”Physical activity and self-rated health during retirement transition – a multi-trajectory analysis of concurrent changes among public sector employees.” (Page 1)

In the introduction, the aim has been rephrased as follows:

“The aim of this study was to evaluate concurrent changes in physical activity and self-rated health during retirement transition among Finnish public sector employees by using a multi-trajectory analysis to identify subgroups with different trajectory paths during this transition.” (Page 8)

We have also discussed generalizability of the findings in the Discussion as follows:

“Although the sample is representative of public sector employees in Finland, the sample was predominated by women (83%) and professional workers (65%), which affects the generalisability of these findings into general population.” (Page 18)

Responses to comments made by reviewer 2:

Comment 1

Please in abstract state the follow up period (generally 2-3 years, I think). Although PA does seem to decline for > 90% of participants, the actual change from peak (in absolute terms) is small. Is it possible to change the message to reflect that the decline is small in the monitored period. Also could abstract state that observed peak seemed to be about 18m post-retirement for most. Without these revisions, the word “temporary” is vague, not defined in the abstract & elsewhere.

Although there is a decline in PA (and it’s good you linked it to self-described health status) for 91%, the decline is actually very small. Surely for public health messaging and ideally supporting people to keep up PA, the rate of decline is important. Calling the increase in PA “temporary” is a bit simplistic because it sounds like a greater decline in PA than was actually observed.

Response 1

The follow-up time was four year and we have now clarified it in the Abstract as follows:

“The aim of the study was to evaluate concurrent changes in physical activity and self-rated health during retirement transition over four years by multivariate trajectory analysis” (Page 4)

The results in the abstract have been rephrased as follows:

“Physical activity peaked at 18 months after retirement and then slightly decreased, except for initially highly physically active participants (9%) with good self-rated health who demonstrated constant high level physical activity .” (Page 4)

Comment 2

Supplemental file 3 has variable vertical scales on charts; trying to same vertical scales on all charts in Supplemental 2 figures makes them all look like flat lines. I would personally prefer either 1) limited scales showing just the range needed or 2) change from baseline which could then be all same scale on Suppl Fig 2 charts. Flat lines are really not informative, though, and don’t help explain the cubic model forms. It could be argued that changes from 3% to 6% developing suboptimal health is really quite a small change in absolute terms, even if it was double.

Response 2

The main results in Figure 2 are shown by using varying vertical scales in order to illustrate changes in physical activity and self-rated health. Supplement 1 figure was added to show the differences in physical activity levels between groups, and thus complements the main findings shown in Figure 2. However, we are ready omit the Supplement 1 figure on the Editor’s request.

In the supplement 2 figure self-rated health variable is treated as a continuous variable. The vertical scale for physical activity is comparable to Figure 2 allowing comparison between these two analytical approaches.

Comment 3

Supplemental files 2/3: are those dashed lines 90% ranges or IQRs or standard deviations ? Please label dashed lines on figures. Definitely did want variance info on the group MET-h.... also, in RESULTS, I think you are talking about “means” not medians as central estimates, with statements

like “moderate level of 22.6” : but not sure. Is that median or mean?

Response 3

The lines are 95% CI:s. This has been updated in the captions of the figures as follows: “The 95% CI:s are presented as dash lines in the figure.”

In the Supplement table 2 we have presented estimated means and their 95% CI:s as labeled in the table. The same estimated means are used in the text.

Comment 4

Please can text reiterate that there were 2.5 years of post-retirement activity data for all participants. This helps to define “temporary”.

Response 4

The following phrase has been added to the methods section:

“The participants received a questionnaire, which has been thereafter sent annually, at least four times, expanding the follow-up 2.5 years after retirement.” (Page 9)

Comment 5

Public sector can cover a wide range of occupational groups. In this study, does public sector include health care workers, teachers, police officers?

Response 5

The following phrase has been added to the text: “Municipalities and cities, involved in the study, are responsible for school education, daycare centers, social services, libraries, maintaining roads, and primary healthcare services among others. Respectively, hospital districts, involved in the study, are responsible for providing secondary healthcare services. Thus, the eligible population covered a wide variety of occupations.” (Page 9)

Comment 6

The abstract suggests that being normal/overweight is associated with higher PA but the detail is that the relevant category is really “not obese”: unless you are saying not a single participant was underweight?

Response 6

The sentence has been rephrased as follows: “Male gender, professional occupation, being married or co-habiting, BMI <30kg/m², not smoking and using alcohol below risk levels were associated with higher physical activity and better self-rated health.” (Page 4)

Comment 7

Sentence that would be more grammatically correct written as “Amounts of > 288 g/week for men and > 192 g/week for women were considered as cut-offs for excess alcohol...”

Response 7

The sentence has been rephrased as suggested. (Page 11)

Comment 8

Penultimate paragraph in methods ends in a sentence (or sentence fragment?) that I didn't understand, twice saying RRs were adjusted for age and gender; is this just unnecessarily repetitive?

Response 8

The sentence has been rephrased as follows: “The RRs were adjusted for age and gender, except for the RR for gender which was only adjusted for age.” (Page 12)

Comment 9

I wasn't sure why mention the module being available for other stats software: It seemed like extraneous detail wrt to this study.

Response 9

The text has now been modified as follows:

"The additional freely available Stata module 'traj' was required to conduct group-based trajectory analysis 29." (Page 12)

Comment 10

Was the achieved sample representative of the eligible respondents? About as female, distributed in about same types of occupations?

Response 10

In the FIREA survey cohort (n=6783), the proportion of women (82% vs. 83%) and professional workers (60% vs. 65%) was comparable to the current study population. Among those who were invited to the FIREA study, but did not respond to any of the surveys, 75% were women and 54% professional workers.

We have commented representativeness of the study population in the Discussion as follows:

"Although the sample is representative of public sector employees in Finland, the sample was predominated by women (83%) and professional workers (65%), which affects the generalisability of these findings into general population." (Page 18)

Comment 11

Group#3 results: text should make clear that there was not a decline from peak PA time after 18m point, unlike declines after 18m point for Group#1 & Group#2.

Response 11

The following phrase has been added: "There was no decline in physical activity, unlike in the other two groups." (Page 14)

Comment 12

Quite often the text uses perfect past tense when simple past tense would be plainer English. For instance, "has reported" could be "reported", "set of original studies was pooled", "participants received a questionnaire which was thereafter sent annually" etc.

Response 12

We have followed the common rules for academic English writing:

- Past perfect for previous research except when the knowledge is absolutely certain
- Past imperfect for the present research
- Present tense for commonly known topics

We are ready to modify the text on the Editor's request.

Comment 13

"compared with those retired due to" would be better "compared with those who retired due to"

Response 13

The sentence has been rephrased as suggested. (Page 8)

Comment 14

.. (double full stop) just before reference 24

Response 14

The typo has been corrected. (Page 10)

Comment 15

Physical activity while commuting is called active travel, conventionally; I wondered if you should use that phrase; leisure PA is a different type of PA

Response 15

In methods, the following sentence has been rephrased: "The participants were asked to estimate their average weekly hours of leisure-time physical activity (including active travel to work) in activities comparable to walking, brisk walking, jogging or running." (Page 10)

In discussion, the following sentence has been rephrased: "This study only assessed leisure-time and active travel to work, leaving out work-related physical activity." (Page 18)

Comment 16

Paragraph about Associations with sociodemographic & Health behavioural factors.... Throughout this paragraph need to say "fewer" not "less"; fewer for countable items.

Response 16

The words "less" have been replaced with "fewer". (Pages 14-15)

Comment 17

I don't think New Zealandian is a word. At least, I think simply "New Zealand" would be better in the adjective form.

Response 17

The word has been corrected as suggested. (Page 17)

Comment 18

Recommend insert "physical" between work-related & activity, penultimate paragr in discussion

Response 18

The word "physical" has been inserted as suggested. (Page 18)

VERSION 2 – REVIEW

REVIEWER	Brainard, Julii University of East Anglia, Norwich Medical School
REVIEW RETURNED	28-Aug-2023
GENERAL COMMENTS	Thanks for addressing my comments.